# SARS-CoV-2 suppresses TLR4-induced immunity by dendritic cells via C-type lectin receptor DC-SIGN

**Lieve E. H. van der Donk**[1,2], **Marta Bermejo-Jambrina**[1,2,3], **John L. van Hamme**[1,2], **Mette M. W. Volkers**[1,2], **Ad C. van Nuenen**[1,2], **Neeltje A. Kootstra**[1,2], **Teunis B. H. Geijtenbeek**[1,2]*

**1** Department of Experimental Immunology, Amsterdam UMC location University of Amsterdam, Amsterdam, The Netherlands, **2** Amsterdam institute for Infection and Immunity, Infectious Diseases, Amsterdam, The Netherlands, **3** Institute of Hygiene and Medical Microbiology, Medical University of Innsbruck, Innsbruck, Austria

* t.b.geijtenbeek@amsterdamumc.nl

## Abstract

SARS-CoV-2 causes COVID-19, an infectious disease with symptoms ranging from a mild cold to severe pneumonia, inflammation, and even death. Although strong inflammatory responses are a major factor in causing morbidity and mortality, superinfections with bacteria during severe COVID-19 often cause pneumonia, bacteremia and sepsis. Aberrant immune responses might underlie increased sensitivity to bacteria during COVID-19 but the mechanisms remain unclear. Here we investigated whether SARS-CoV-2 directly suppresses immune responses to bacteria. We studied the functionality of human dendritic cells (DCs) towards a variety of bacterial triggers after exposure to SARS-CoV-2 Spike (S) protein and SARS-CoV-2 primary isolate (hCoV-19/Italy). Notably, pre-exposure of DCs to either SARS-CoV-2 S protein or a SARS-CoV-2 isolate led to reduced type I interferon (IFN) and cytokine responses in response to Toll-like receptor (TLR)4 agonist lipopolysaccharide (LPS), whereas other TLR agonists were not affected. SARS-CoV-2 S protein interacted with the C-type lectin receptor DC-SIGN and, notably, blocking DC-SIGN with antibodies restored type I IFN and cytokine responses to LPS. Moreover, blocking the kinase Raf-1 by a small molecule inhibitor restored immune responses to LPS. These results suggest that SARS-CoV-2 modulates DC function upon TLR4 triggering via DC-SIGN-induced Raf-1 pathway. These data imply that SARS-CoV-2 actively suppresses DC function via DC-SIGN, which might account for the higher mortality rates observed in patients with COVID-19 and bacterial superinfections.

## Author summary

Superinfections with bacteria during severe COVID-19 lead to worse prognosis of COVID-19 patients. Here we have identified a novel mechanism triggered by SARS-CoV-2 that suppresses immune responses to bacteria. We found that exposure of human dendritic cells (DCs) to SARS-CoV-2 or the Spike glycoprotein blocks TLR4-induced immune

**Data Availability Statement:** The numerical data used in all figures are included in S1 Data.

**Funding:** This research was funded by the Netherlands Organisation for Health Research and Development together with the Stichting Proefdiervrij (ZonMW MKMD COVID-19 grant nr. 114025008 to TBHG) and European Research Council (Advanced grant 670424 to TBHG). LEHvdD was supported by the Netherlands Organization for Scientific Research (NWO) (Grant number: 91717305). MBJ was supported by a Work Visit Grant of the Amsterdam institute for Infection and Immunity and by an APART-MINT Fellowship of the Austrian Academy of Sciences at the Institute of Hygiene and Medical Microbiology of the University of Innsbruck (Grant number 11978). The funders had no role in study design, data collection and analysis, decision to publish, or preparation of the manuscript.

**Competing interests:** The authors have declared that no competing interests exist.

responses. Our data strongly suggest that SARS-CoV-2 interacts with the C-type lectin receptor DC-SIGN, which blocks TLR4-induced immunity. Taken together, our data suggest that SARS-CoV-2 actively suppresses DC function via DC-SIGN, which might account for the higher mortality rates observed in patients with COVID-19 and bacterial superinfections.

## Introduction

SARS-CoV-2 causes coronavirus disease 2019 (COVID-19), which is an infectious disease characterized by strong induction of inflammatory cytokines, progressive lung inflammation and potentially multi-organ dysfunction [1–3]. SARS-CoV-2 infects epithelial cells of the airways using the receptor angiotensin-converting enzyme 2 (ACE2) for infection [4,5]. Notably, it has been reported that COVID-19 patients, in particular severely ill patients, are vulnerable to viral, fungal or bacterial superinfections [6–10]. Superinfections arise when a primary infection is followed by a secondary infection [11]. Many of the superinfections in COVID-19 patients are caused by bacteria, for instance through hospital-acquired pneumonia or ventilator-acquired pneumonia with different virulent bacterial species such as *Pseudomonas (P.) aeruginosa* or *Klebsiella (K.) pneumoniae* [6–10,12]. Severe COVID-19 with superinfections is therefore associated with significantly worse prognosis [10,12]. However, it is currently unclear whether increased susceptibility of COVID-19 patients to bacterial superinfections is due to systemic inflammation or SARS-CoV-2 specifically affecting defense against bacterial infections.

Dendritic cells (DCs) are located throughout the mucosal barrier tissues such as the airways and lungs, and are essential for defense against infections by microbes including bacteria and viruses. DCs sense foreign microbes with pattern recognition receptors (PRRs), leading to antigen presentation to T cells and potent adaptive immune responses [13]. Toll-like receptors (TLR) are important PRRs for sensing bacteria and TLR triggering induces type I Interferon (IFN) and proinflammatory cytokine responses, required for adaptive immunity [14]. The TLR family member TLR4 is highly expressed by DCs and senses the bacterial component lipopolysaccharide (LPS) [15]. Notably, whereas recent research has focused on TLR4-mediated immune activation by SARS-CoV-2 [16–19], the suppressive effects of SARS-CoV-2 on the immune response are not yet investigated.

Here, we investigated whether SARS-CoV-2 affects DC-induced immunity to bacteria using both SARS-CoV-2 S protein and SARS-CoV-2 primary isolate (hCoV-19/Italy). Notably, our data strongly suggest that SARS-CoV-2 suppresses DC-induced immune responses by TLR4, whilst signaling through other TLRs was not affected. Our data suggest that SARS-CoV-2 S protein as well as SARS-CoV-2 virus particles bind DC-SIGN, which induces signaling via kinase Raf-1 suppressing TLR4 signaling. Thus, we have identified a novel mechanism of immunosuppression by SARS-CoV-2 that might underlie the increased susceptibility to Gram-negative bacteria and targeting this pathway might attenuate bacterial infections during COVID-19.

## Results

### SARS-CoV-2 S protein specifically suppresses TLR4 activation

To investigate whether SARS-CoV-2 affects DC function towards other external stimuli, we exposed DCs to recombinant SARS-CoV-2 S protein before adding different TLR agonists, and screened for immune responses by measuring induction of interferon (IFN)-stimulated

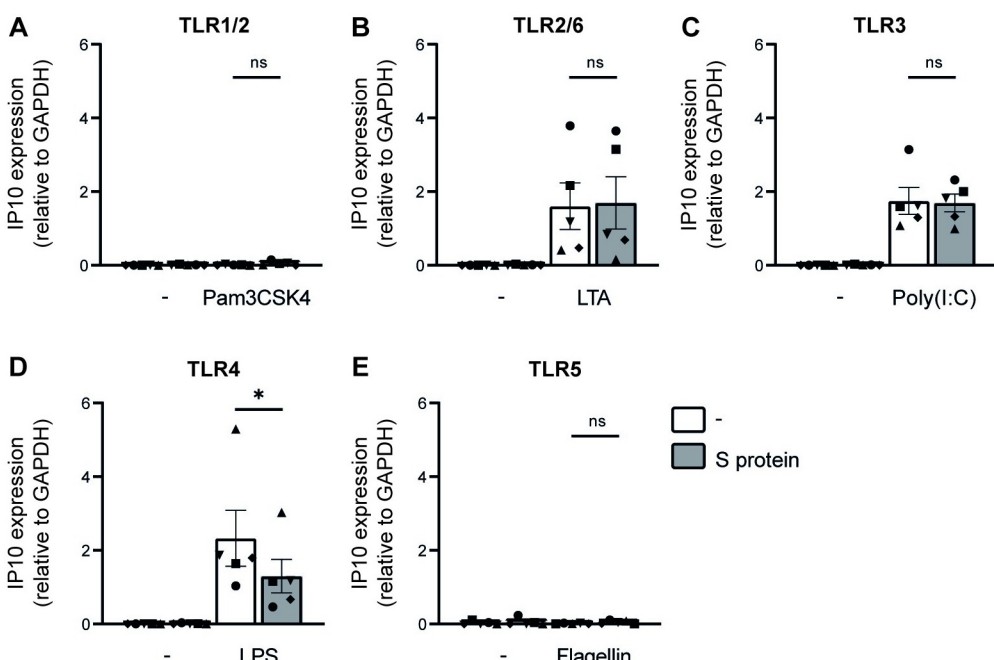

**Fig 1. S protein modulates TLR4 signaling.** (A-E) DCs were pre-incubated with S protein before exposure to a plethora of bacterial TLR stimuli. After 6h incubation, cells were lysed and mRNA levels of IP10 were determined by qPCR. Data show the mean values and SEM. Statistical analysis was performed using student's *t*-test. Data represent n = 5 DC donors obtained in three separate experiments with each symbol representing a different donor. *p<0.05; ns = non-significant.

gene (ISG) IP10. As we have previously shown, S protein alone did not induce DC activation [19]. TLR agonists against TLR2/6, TLR3 and TLR4 induced IP10 (Fig 1), while TLR1/2 and TLR5 agonists did not. Notably, pre-incubation with S protein decreased induction of IP10 by the TLR4 agonist, but not by the other TLR agonists. These data suggest that SARS-CoV-2 specifically modulates TLR4 signaling.

## SARS-CoV-2 S protein is involved in the suppression of immune responses by DCs

We next determined the effect of SARS-CoV-2 on TLR4-induced immune responses. DCs from healthy donors were exposed to recombinant S protein before adding TLR4 agonist lipopolysaccharide (LPS) and type I IFN and cytokine responses were determined. LPS alone induced mRNA levels of IFN-β and ISGs IP10 and ISG15, and cytokines interleukin (IL)-6 and IL-10 (Fig 2A–2E). Notably, pre-exposure to recombinant S protein significantly reduced mRNA levels of IFN-β, IP10 and ISG15 as well as IL-6 and IL-10. As published before, S protein alone did not induce any type I IFN or cytokine responses [19]. Our data therefore strongly suggest that SARS-CoV-2 S protein suppresses both TLR4-induced type I IFN and cytokine responses.

## SARS-CoV-2 S protein does not directly affect TLR4 signaling

SARS-CoV-2 has been suggested to interact with TLR4 [17,20,21] and therefore we investigated whether S protein could sterically hinder the binding of LPS using a TLR4-expressing HEK293 cell line (HEK293/TLR4). HEK293 cells do not inherently express TLRs or DNA sensors, but

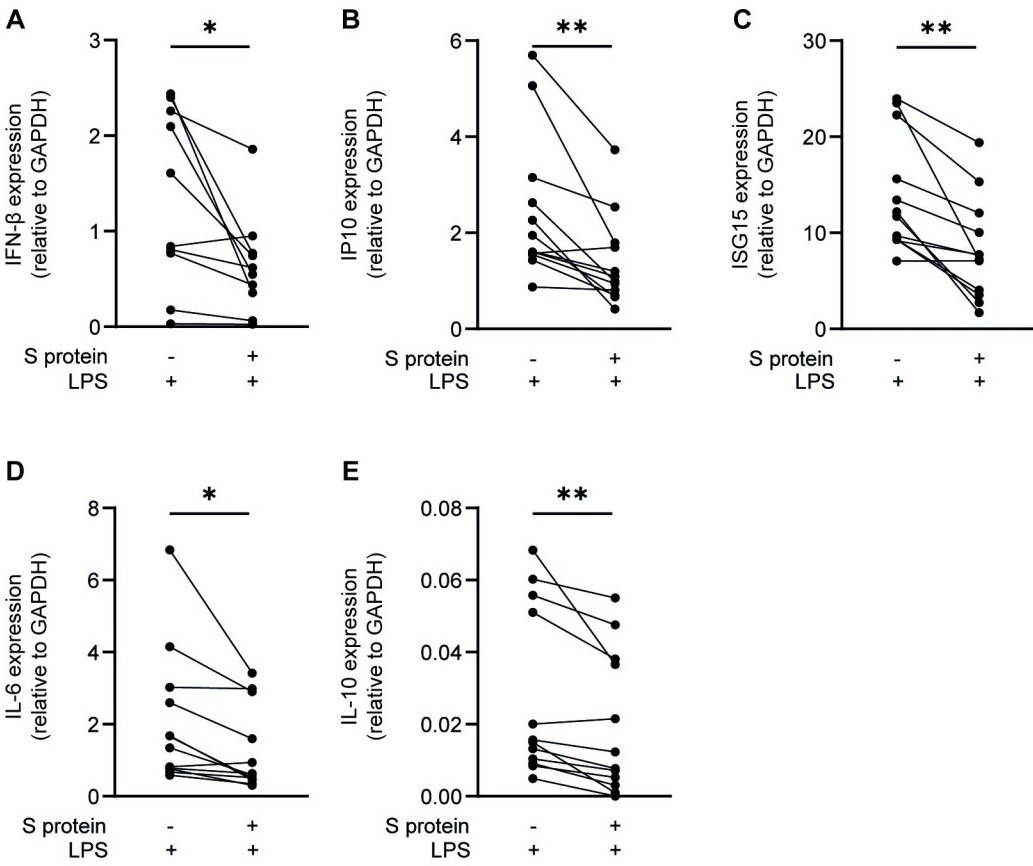

**Fig 2. S protein modulates TLR4-mediated immune responses by DCs.** (A-E) DCs were exposed to S protein prior to addition of TLR4 agonist LPS. After 2h or 6h incubation, cells were lysed and mRNA levels of IFN-β (A) were determined after 2h, and mRNA expression levels of IP10 (B), ISG15 (C) and cytokines IL-6 (D) and IL-10 (E) were determined after 6h by qPCR. Data show expression for n = 10 donors obtained in 5 separate experiments (2h stimulation) or n = 12 donors obtained in 6 separate experiments (6h stimulation). Statistical analysis was performed using student's *t*-test. \*\*p<0.01; \*p<0.05.

ectopic expression and triggering of TLR4 leads to the production of the cytokine IL-8. In contrast to parental HEK293 cells, incubation of HEK293/TLR4 cells with LPS induced IL-8 production (Fig 3A and 3B) [19]. Pre-incubation with a low and high concentration of recombinant S protein did not affect LPS-induced IL-8 expression by the HEK293/TLR4 cells (Fig 3A and 3B). Additionally, pre-incubation of HEK293/TLR4 cells with SARS-CoV-2 primary isolate (hCoV-19/Italy) prior to exposure to LPS did also not affect IL-8 production (Fig 3C). Whilst IL-8 is the primary readout in this assay, we aimed to confirm our findings by determining other cytokines. TLR4-mediated IFN responses were not detected, but the inflammatory cytokine TNF-α was induced upon TLR4 triggering, and remained unaffected by pre-incubation with S protein or SARS-CoV-2 primary isolate (Fig 3D–3F). Taken together, these results suggest that neither S protein nor SARS-CoV-2 primary isolate directly suppress TLR4 signaling either by sterically hindering binding of LPS to TLR4 or direct modulation of TLR4 signaling.

## SARS-CoV-2 S protein binds DC-SIGN expressed by DCs

Next we investigated whether crosstalk with the C-type lectin receptor (CLR) DC-SIGN is involved in the modulation of TLR4 signaling. Previous reports with different pathogens have

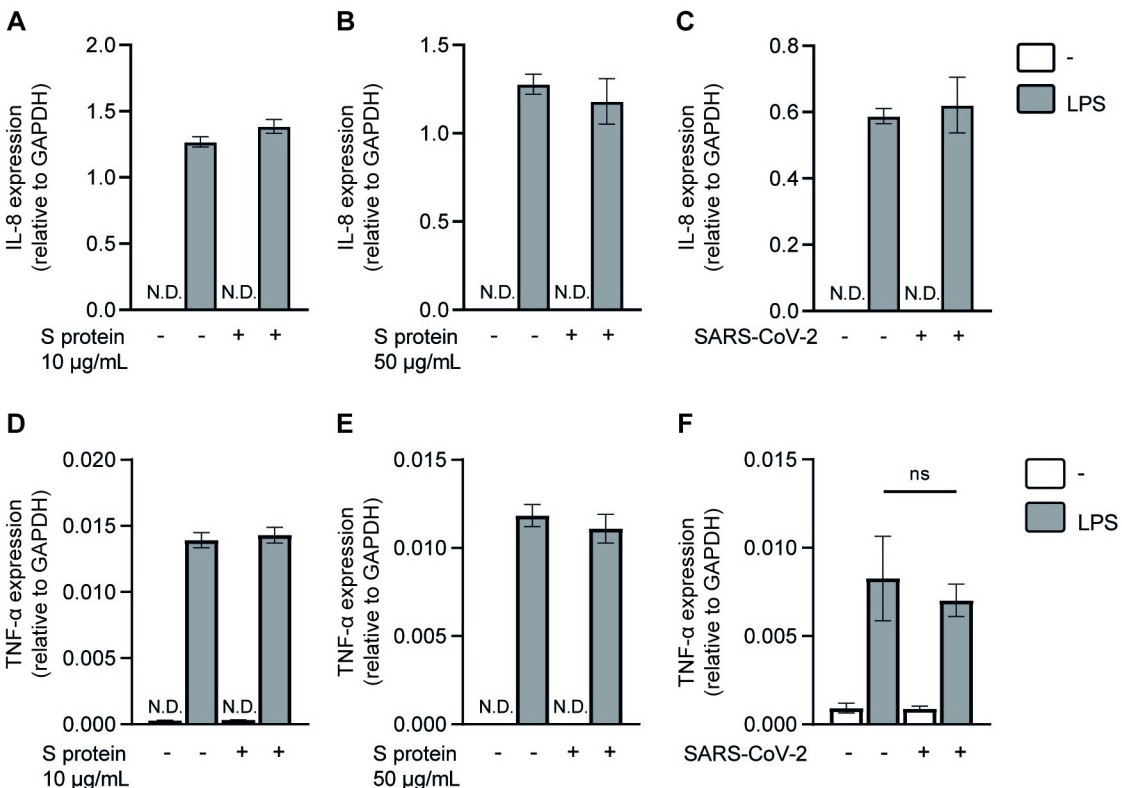

**Fig 3. S protein or SARS-CoV-2 primary isolate does not sterically hinder binding of LPS to TLR4.** (A-F) HEK293/TLR4 cells were pre-incubated for 2h with a low (A, D) or high (B, E) concentration of S protein or SARS-CoV-2 primary isolate (C, F) before exposure to TLR4 agonist LPS. After 24h incubation, cells were lysed and mRNA levels of IL-8 (A-C) and TNF-α (D-F) were determined by qPCR. Data show the mean values and SEM obtained in three separate experiments. Statistical analysis was performed using one-way ANOVA. ns = non-significant; N.D. = not detected.

shown that DC-SIGN signaling modulates immune responses by DCs [22–25] and SARS-CoV-2 S protein has been shown to interact with DC-SIGN [26,27]. However, whilst these reports show SARS-CoV-2 S protein binding to DC-SIGN-overexpressing cell lines, the direct binding of S protein to DC-SIGN expressed by primary DCs has not yet been elucidated. Therefore we investigated whether human DCs interact with SARS-CoV-2 S protein via DC-SIGN using a S-protein-labeled fluorescent bead binding assay [28]. Notably, S protein strongly bound to DCs and binding was abrogated by the CLR inhibitor mannan and blocking antibodies against DC-SIGN (Fig 4A). Moreover, DCs efficiently captured SARS-CoV-2 virus particles and binding was blocked by mannan as well as anti-DC-SIGN antibodies (Fig 4B). Isotype controls did neither affect the binding of S protein nor SARS-CoV-2 virus particles to DCs. These data strongly suggest that SARS-CoV-2 binds DC-SIGN on primary DCs via envelope glycoprotein S.

## SARS-CoV-2 suppresses TLR4-mediated DC activation via DC-SIGN

Next we investigated whether SARS-CoV-2 suppresses TLR4 functionality via DC-SIGN. DCs from healthy donors were treated with recombinant SARS-CoV-2 S protein prior to LPS stimulation in presence or absence of antibodies against DC-SIGN, and type I IFN and cytokine responses were determined. Exposure to S protein decreased LPS-induced mRNA levels of IFN-β (p = ns), IP10 (p = 0.017) and ISG15 (p = 0.0059) (Fig 5A–5C). Antibodies against

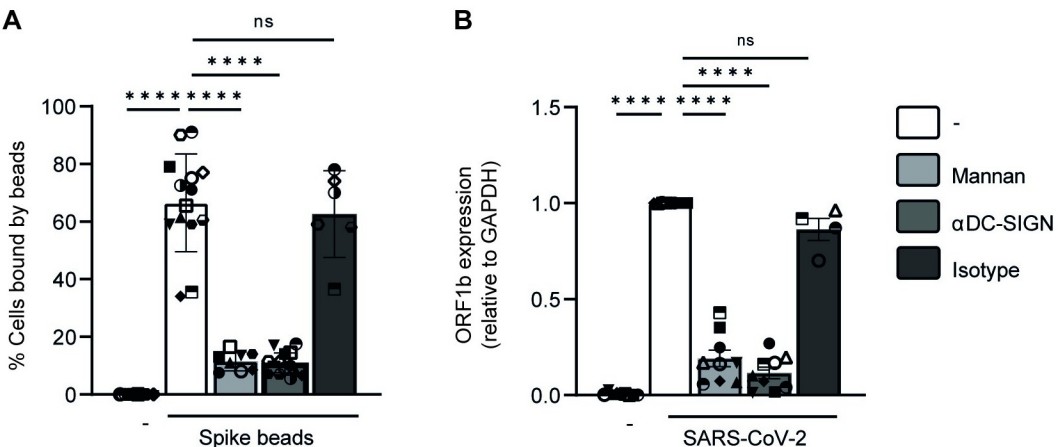

**Fig 4. S protein binds DC-SIGN expressed by DCs.** (A) DCs were exposed to S-protein-coated fluorescent beads in the absence or presence of CLR block mannan, DC-SIGN blocking antibodies, or an isotype control after which S protein binding to DCs was determined by flow cytometry. (B) DCs were exposed to SARS-CoV-2 primary isolate in the absence or presence of CLR block mannan, DC-SIGN blocking antibodies, or an isotype control after which virus binding to DCs was determined by qPCR. SARS-CoV-2 binding was set at 1 in cells treated with SARS-CoV-2 primary isolate. Data show the mean values and SEM. Statistical analysis was performed using one-way ANOVA. Data represent n = 8–14 donors obtained in 6 separate experiments performed in triplo (A) or n = 4–9 donors obtained in 5 separate experiments performed in triplo (B) with each symbol representing a different donor. ****$p < 0.0001$; ns = non-significant.

DC-SIGN restored IP10 (p = ns) and ISG15 (p = 0.025) expression to levels observed with LPS alone (Fig 5A–5C). IFN-β expression was not affected by antibodies against DC-SIGN. Similarly, exposure to S protein decreased LPS-induced mRNA levels of IL-6 (p = ns), and IL-10 (p = 0.042) (Fig 5D and 5E). S protein-mediated suppression of IL-6 and IL-10 was restored by blocking DC-SIGN, albeit not significantly and varied amongst DC donors (Fig 5D and 5E). These data suggest that DC-SIGN binding by S protein suppresses TLR4 signaling.

Next we investigated whether SARS-CoV-2 primary isolate suppresses LPS-induced immune responses by DCs. Previously we have shown that SARS-CoV-2 primary isolate does not induce type I IFN and cytokine responses by DCs [19]. Strikingly, pre-incubation of DCs with SARS-CoV-2 primary isolate prior to exposure to LPS suppressed mRNA levels of IFN-β (p = ns), ISGs IP10 (p = 0.045) and ISG15 (p = ns), as well as cytokines IL-6 (p = 0.014) and IL-10 (p = ns) (Fig 6A–6E). Moreover, anti-DC-SIGN antibodies restored expression of type I IFN and cytokine responses to levels observed with LPS alone (Fig 6A–6E), although this effect was only significant for IP10 (p = 0.02) and IL-6 (p = 0.02). DC-SIGN signaling via mannose-expressing pathogens triggers Raf-1 activation leading to immune modulation [29]. We therefore studied whether a small molecule inhibitor of Raf-1 (GW5074) affects SARS-CoV-2-suppression of LPS signaling. Although Raf-1 inhibition did not affect SARS-CoV-2 suppression of IFN-β (p = ns), the expression of ISGs IP10 (p = 0.02) and ISG15 (p = ns) as well as cytokines IL-6 (p = 0.03) and IL-10 (p = ns) were restored by inhibiting Raf-1 to levels observed for LPS alone (Fig 6A–6E). The differences in significance are likely due to donor variability. These results suggest that SARS-CoV-2 modulates DC activation through DC-SIGN, thereby disabling DCs to respond to bacterial superinfections during COVID-19.

## Discussion

The SARS-CoV-2 pandemic has made an enormous impact all over the world. The high morbidity and mortality rates are not merely due to SARS-CoV-2 infection and aberrant immune

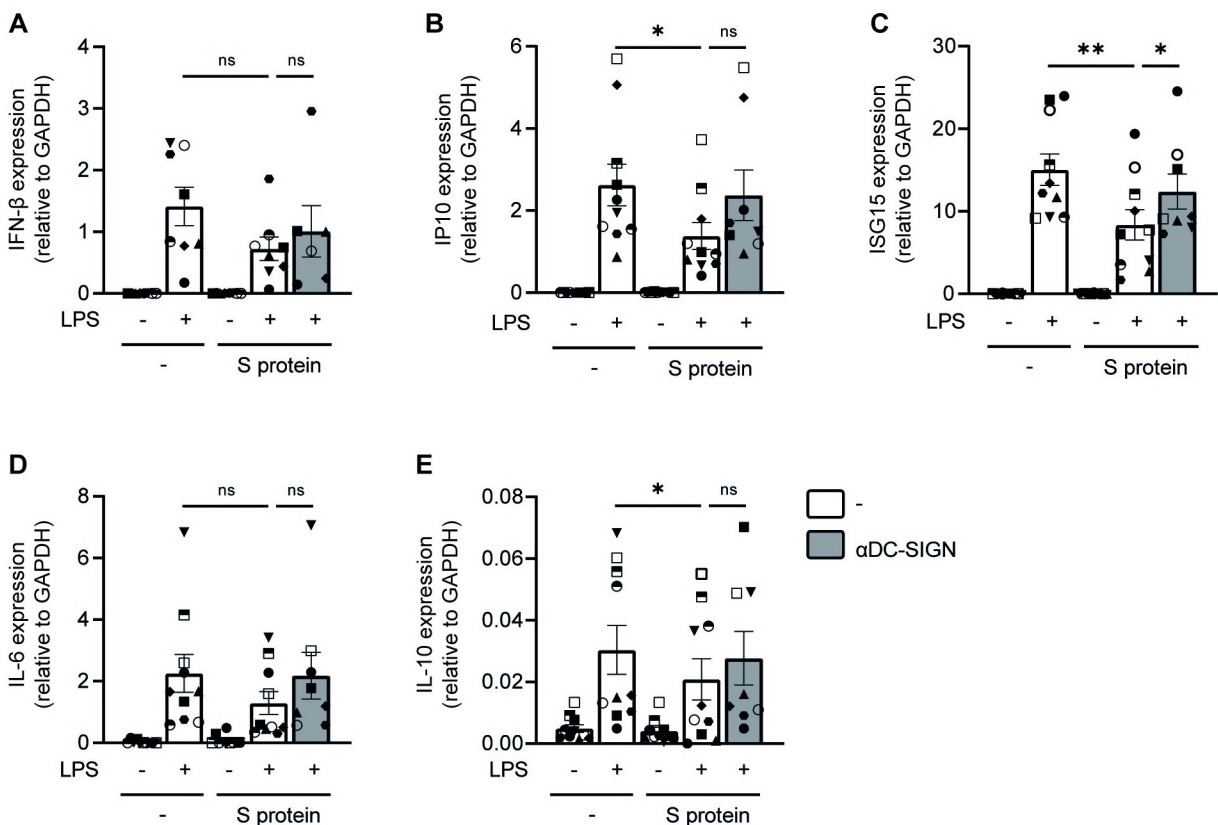

**Fig 5. S protein suppresses TLR4-induced immunity via DC-SIGN.** (A-B) DCs were incubated with S protein in the absence or presence of anti-DC-SIGN blocking antibodies before exposure to LPS. After 2h or 6h incubation, DCs were lysed and mRNA transcription of IFN-β (A), IP10 (B), ISG15 (C), IL-6 (D) and IL-10 (E) was determined by qPCR. Data show the mean values and SEM. Statistical analysis was performed using one-way ANOVA with Tukey's multiple comparison's test for mixed-effects analysis. Data represent n = 6–8 donors obtained in 4 separate experiments (2h stimulation) or n = 8–10 donors obtained in 6 separate experiments (6h stimulation) with each symbol representing a different donor. **p<0.01; *p<0.05; ns = non-significant.

responses against the virus, but are also due to superinfections [6–12]. Hospitalized patients with severe COVID-19 are susceptible to superinfections with other viruses, fungi or bacteria. Often these superinfections are caused by bacteria leading to pneumonia, bacteremia and sepsis [30,31]. Damage inflicted on lung tissue by SARS-CoV-2, and mechanical stress caused by intubations are major reasons for the spread of certain bacteria in the lungs and throughout the body [9,30,31]. However, other mechanisms might underlie increased susceptibility to bacterial superinfections, such as decreased function of DCs during COVID-19 [32]. Here we have identified a novel pathway activated by SARS-CoV-2 that suppresses TLR4, the major bacterial TLR on human DCs. Our data strongly suggest that SARS-CoV-2 interacts with the C-type lectin receptor DC-SIGN leading to Raf-1-mediated suppression of TLR4 signaling. We observed suppression of both type I IFN and cytokine responses and these inflammatory mediators are crucial in the defense against bacterial infections [33,34].

Bacterial superinfections are caused by both Gram-positive bacteria, including *Streptococcus pneumoniae* and Gram-negative bacteria including *P. aeruginosa* and *K. pneumoniae* [12]. Notably, DCs responded similarly to stimulation with Gram-positive stimuli Pam3CSK4 and LTA, irrespective of pre-exposure to recombinant SARS-CoV-2 S protein. Moreover, DCs in presence or absence of S protein also reacted similarly to the Gram-negative stimulus flagellin,

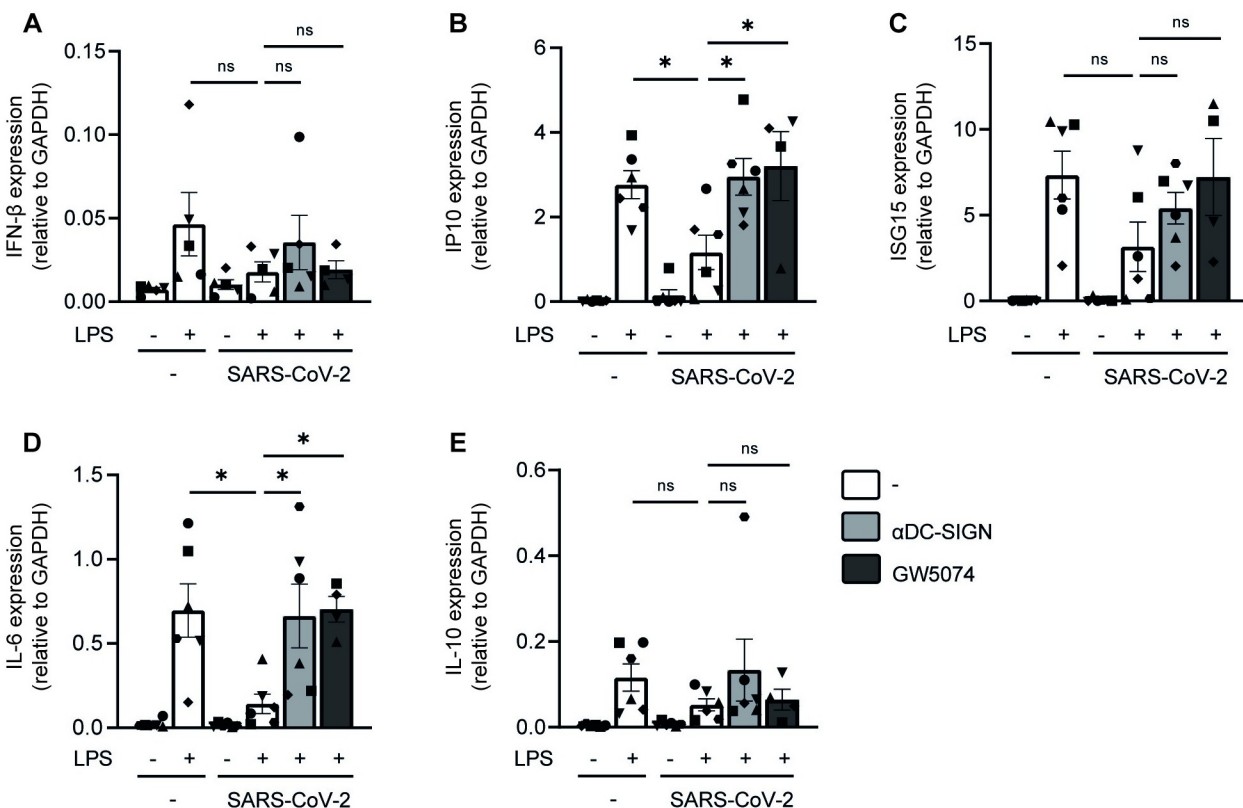

**Fig 6. SARS-CoV-2 primary isolate suppresses DC immunity via DC-SIGN.** (A-B) DCs were incubated with SARS-CoV-2 primary isolate in the absence or presence of anti-DC-SIGN blocking antibodies before exposure to LPS. After 2h or 6h incubation, DCs were lysed and mRNA transcription of IFN-β (A), IP10 (B), ISG15 (C), IL-6 (D) and IL-10 (E) was determined by qCPR. Data show the mean values and SEM of n = 4–5 donors obtained in 3 separate experiments (2h stimulation) or n = 4–6 donors obtained in 3 separate experiments (6h stimulation) with each symbol representing a different donor. Statistical analysis was performed using one-way ANOVA with Tukey's multiple comparison's test. *p<0.05; ns = non-significant.

whereas DCs were significantly less responsive towards LPS in the presence of S protein. These findings led us to investigate how TLR4 binding or signaling was affected. Another study reported decreased functioning of DCs from COVID-19 patients towards TLR triggers during acute SARS-CoV-2 infection as DC activation upon TLR3, TLR4, TLR7 and TLR8 triggering was suppressed [35]. Here we observed that SARS-CoV-2 specifically affected TLR4 signaling. As we have used DCs from healthy donors, our data suggest that SARS-CoV-2 as well as S protein directly affects TLR4 signaling.

Some pathogens are known to mimic pathogen or host structures to remain hidden from immune detection or become more pathogenic by inducing alternative signaling [36–40]. Previous research suggests that S protein binds TLR4 to trigger immune activation [17,20,21, 41]. We have shown that S protein does not activate DCs through TLR4 triggering [19]; however, S protein might still bind TLR4 and thereby inhibit TLR4 signaling. TLR4 activation in HEK293 cells ectopically expressing TLR4 was neither affected by recombinant SARS-CoV-2 S protein nor SARS-CoV-2 infectious virus. These data strongly suggest that SARS-CoV-2 does neither sterically block binding of LPS to TLR4 nor directly inhibit TLR4 signaling. Besides TLRs, DCs express many different PRRs involved in virus binding. An important PRR family are the CLRs that interact with pathogens via carbohydrates and have been shown to induce signaling that directs or modulates immune responses [42,43]. In particular, the CLR DC-SIGN is

expressed by DCs and macrophages, and recognizes high-mannose-containing glycoproteins on the surface of pathogens [44,45]. Previous research has shown that DC-SIGN modulates immune activation towards various pathogens [25,45,46]. ManLAM, a highly mannosylated cell-wall component of *Mycobacterium (M.) tuberculosis*, interacts with DC-SIGN resulting in an altered immune response through crosstalk between DC-SIGN and TLR4 [22,25]. Interaction between ManLAM and DC-SIGN modulates TLR4 signaling via Raf-1 resulting in phosphorylation and acetylation of NFκB, which enhances induction of IL-10, IL-12 and IL-6 [22,25]. Similarly, envelope glycoprotein of HIV-1 enhances LPS-induced IL-10, IL-12 and IL-6 responses via DC-SIGN [29]. However, our data strongly suggest that SARS-CoV-2 interaction with DC-SIGN suppresses TLR4 signaling via Raf-1. Thus, DC-SIGN modulation of immunity is strongly dependent on the PRR triggered as well as the pathogen that is recognized by DC-SIGN, which might thereby underlie the observed differences between SARS-CoV-2 and HIV-1. It is currently unclear what causes the different effects on TLR4 signaling induced by HIV-1 or SARS-CoV-2 on DCs. DC-SIGN triggering strongly depends on the sugar moieties expressed on the viral envelope [46]. Whilst both HIV-1 and SARS-CoV-2 express highly mannosylated envelope proteins, the HIV-1 envelope glycoprotein gp120 is mostly composed of oligomannose structures [47], whereas the SARS-CoV-2 envelope S glycoprotein mainly includes complex glycosylation [48,49]. In addition, glycoprotein density between HIV-1 and SARS-CoV-2 virions might account for differences as density of glycoproteins on virions might lead to different receptor crosslinking and signaling [50,51]. These differences in the envelope glycoproteins might explain the different downstream effects of DC-SIGN signaling. Moreover, DCs can be infected by HIV-1, whilst DCs are not infected by SARS-CoV-2. Possibly internalization of HIV-1 in the DC's endosomal compartment induces complementary intracellular signaling that is not induced by SARS-CoV-2. However, for SARS-CoV-2 we can also not exclude involvement of other receptors. Further research on the effect of DC-SIGN triggering during SARS-CoV-2 infection is required to elucidate underlying mechanisms. In addition, it is important to note that also coinfections or superinfections with other viruses and fungi were reported [6,12,52–54]. It would be interesting to further investigate how superinfections with different bacteria, viruses or fungi might affect DC function during COVID-19. In conclusion, our data suggest that SARS-CoV-2 S protein-mediated DC-SIGN crosstalk affects TLR4-induced immunity, which might underlie bacterial superinfections during COVID-19.

## Materials and methods

### Ethics approval statement

This study was performed in accordance with the ethical principles set out in the declaration of Helsinki and was approved by the institutional review board of the Amsterdam University Medical Centers, location AMC Medical Ethics Committee and the Ethics Advisory Body of Sanquin Blood Supply Foundation (Amsterdam, Netherlands). Written informed consent was obtained from all participants.

### Cell lines

The Simian kidney cell line VeroE6 (ATCC CRL-1586) was maintained in $CO_2$-independent medium (Gibco Life Technologies, Gaithersburg, Md.) supplemented with 10% fetal calf serum (FCS), 2mM L-glutamine and penicillin/streptomycin (Invitrogen). Cultures were maintained at 37˚C without $CO_2$. The human embryonic kidney (HEK) 293 cells (ATCC CRL-11268) were maintained in Iscove's modified Dulbecco's medium (IMDM) (Gibco Life Technologies) containing 10% FCS, L-glutamine, and 1% penicillin/streptomycin. Cultures

were maintained at 37°C and 5% $CO_2$. HEK293 cells stably transfected with TLR4 cDNA (HEK/TLR4) were a kind gift from Dr. T. Golenbock [15]. Cells were split and seeded into flat-bottom 96-well plates (Corning) and left to attach for 24h, before performing further experiments.

## Primary cells

This study was performed in accordance with the ethical principles set out in the declaration of Helsinki and was approved by the institutional review board of the Amsterdam University Medical Centers, location AMC Medical Ethics Committee and the Ethics Advisory Body of Sanquin Blood Supply Foundation (Amsterdam, Netherlands). Written informed consent was obtained from all participants. Human CD14+ monocytes were isolated from the blood of healthy volunteer donors and subsequently differentiated into monocyte-derived dendritic cells (DCs) as described before [24]. In short, the isolation of monocytes from buffy coats was performed by density centrifugation on Lymphoprep (Nycomed) and Percoll (Pharmacia). Monocytes were cultured in Roswell Park Memorial Institute (RPMI) 1640 (Gibco), supplemented with 10% FCS, 2mM L-glutamin (Invitrogen), and 1% penicillin/streptomycin. Differentiation into DCs was performed by the addition of cytokines IL-4 (500U/mL) and GM-CSF (800U/mL) (both Gibco). After 4 days of differentiation, DCs were seeded at 1 x 10^6/mL in a round-bottom 96-well plate (Corning). After 2 days of recovery, DCs were stimulated as described below.

## SARS-CoV-2 (hCoV-19/Italy) virus production

The following reagent was obtained from Dr. Maria R. Capobianchi through BEI Resources, NIAID, NIH:SARS-related coronavirus 2, Isolate Italy-INMI1, NR-52284, originally isolated in January 2020 in Rome, Italy. SARS-CoV-2 virus productions were performed as described before [19,55] In brief, VeroE6 cells were inoculated with the SARS-CoV-2 primary isolate and incubated for 48h, after which virus supernatant was harvested. Tissue culture infectious dose (TCID50) was determined on VeroE6 cells by MTT assay. MTT staining is indicative of cell viability and can be measured using a spectrometer. The virus titer was determined as TCID50/mL and calculated based on the Reed Muench method [56] as described before [55].

## Reagents and stimulations

DCs were left unstimulated or exposed to 10 μg/mL SARS-CoV-2 S protein (Bio-techne) for 1h, after which DCs were exposed to the following TLR stimuli: 10μg/mL Pam3CSK4 (Invivogen), 10 μg/mL Poly(I:C) (Invivogen), 10 ng/mL LPS from *Salmonella typhi* (Sigma), 10 μg/mL flagellin from *Bacillus subtilis* (Invivogen), 10 μg/mL lipotechoic acid from *Staphylococcus aureus* (Invivogen). To investigate the contribution of DC-SIGN and Raf-1, cells were pre-incubated with 20μg/mL anti-DC-SIGN blocking antibody AZN-D1 for 30 min, or with 1μM GW5074 (Calbiochem) for 2h, respectively, before adding S protein. For exposure to SARS-CoV-2 (hCoV-19/Italy), DCs were incubated with inhibitors prior to exposure to SARS-CoV-2 TCID1000 for 1h, and then to LPS for 2 or 6h, after which cells were lysed.

Similarly, HEK293 and HEK/TLR4 cells were incubated for 2h with SARS-CoV-2 S protein or SARS-CoV-2 (hCoV-19/Italy), after which LPS was added. Cells were lysed after 24h for qPCR analysis.

## RNA isolation and quantitative real-time PCR

Cells exposed to SARS-CoV-2 primary isolate (hCoV-19/Italy) were lysed and RNA was isolated with the QIAamp Viral RNA Mini Kit (Qiagen) according to the manufacturer's protocol. cDNA was synthesized using the M-MLV reverse-transcriptase kit (Promega). Before further application, cDNA was diluted 1 in 5 in depc. Cells exposed to SARS-CoV-2 S protein were lysed and RNA was isolated with the RNA Catcher PLUS kit (Invivogen) according to the manufacturer's instructions. Subsequently, cDNA was synthesized with a reverse-transcriptase kit (Promega). PCR amplification was performed in the presence of SYBR green (Thermofisher) in a 7500 Fast Realtime PCR system (ABI). Specific primers were designed using Primer Express 2.0 (Applied Biosystems). The comparative delta Ct method was used to normalize the amount of target mRNA to the expression of household gene GAPDH.

The following primers were used:

GAPDH: F_CCATGTTCGTCATGGGTGTG; R_GGTGCTAAGCAGTTGGTGGTG; IFNB: F_ACAGACTTACAGGTTACCTCCGAAAC; R_CATCTGCTGGTTGAAGAATGCTT; ISG15: F_ TTTGCCAGTACAGGAGCTTGTG; R_ GGGTGATCTGCGCCTTCA; CXCL10: F_ CGCTGTACCTGCATCAGCAT; R_ CATCTCTTCT-CACCCTTCTTTTTCA; IL-6: F_TGCAATAACCACCCCTGACC; R_TGCGCAGAAT-GAGATGAGTTG; IL-10: F_GAGGCTACGGCGCTGTCAT; R_CCACGGCCTTGCTCTTGTT; ORF1b: F_TGGGGTTTTACAGGTAACCT; R_AACACGCTTAACAAAGCACT; TNF: F_ CCAAGCCCTGGTATGAGCC; R_ GCCGATTGATCTCAGCGC.

## Bead binding and SARS-CoV-2 isolate (hCoV-19/Italy) binding assays

To investigate ligand-receptor interactions, we used a fluorescent bead binding assay as described before [28]. PerCP fluorescent streptavidin beads were coated with biotinylated S protein (Bio-techne). DCs were seeded at a density of 50.000 cell/well in a 96-well V-bottom plate in TSA (TSA: 0.5% bovine serum albumin (BSA) in TSM (200mM Tris, 1500mM NaCl, 10mM $CaCl_2$, 20mM $MgCl_2$) pH 7.4). Subsequently, cells were incubated with TSA, 20µg/mL anti-DC-SIGN antibody AZN-D1, 100µg/mL mannan, or 20µg/mL anti-langerin antibody isotype 10E2 for 30 min at 37˚C. Beads were added to each corresponding well in a 1:20 dilution and incubated at 37˚C for 45 min. After washing once with TSA, cells were resuspended in TSA and adhesion was measured on a FACS Canto flow cytometer (BD Biosciences). The data was analyzed using FlowJo V10 software (Treestar).

To assess virus binding, DCs were exposed to 20µg/mL anti-DC-SIGN blocking antibody AZN-D1, 50µg/mL mannan, or 20µg/mL anti-langerin antibody isotype 10E2 for 30 min at 37˚C prior to incubation with SARS-CoV-2 isolate (hCoV-19/Italy) for 2h at 4˚C. After 2h, cells were washed extensively with phosphate-buffered saline (PBS) to remove unbound virus and subsequently lysed with AVL buffer (Qiagen). RNA and cDNA were prepared as described above, and the amount of virus bound was determined with qPCR using ORF1b primers [57].

## Statistics

Graphpad Prism version 8 (GraphPad Software) was used to generate all graphs and to perform statistical analyses. For pairwise comparisons, a Student's *t*-test was used. Multiple comparisons within groups were performed using a one-way ANOVA with a Tukey's multiple comparisons test, or two-way ANOVA with a Tukey's multiple comparisons test, where indicated. $p < 0.05$ were considered statistically significant.

## Supporting information

**S1 Data. Excel spreadsheet containing the underlying numerical data for Figs 1–6 in separate sheets.**
(XLSX)

## Author Contributions

**Conceptualization:** Lieve E. H. van der Donk, Marta Bermejo-Jambrina, Teunis B. H. Geijtenbeek.

**Data curation:** Lieve E. H. van der Donk, Marta Bermejo-Jambrina, John L. van Hamme, Mette M. W. Volkers, Ad C. van Nuenen.

**Formal analysis:** Lieve E. H. van der Donk.

**Funding acquisition:** Marta Bermejo-Jambrina, Teunis B. H. Geijtenbeek.

**Investigation:** Lieve E. H. van der Donk, Marta Bermejo-Jambrina, John L. van Hamme, Mette M. W. Volkers, Ad C. van Nuenen, Neeltje A. Kootstra, Teunis B. H. Geijtenbeek.

**Methodology:** Lieve E. H. van der Donk, Marta Bermejo-Jambrina, John L. van Hamme, Mette M. W. Volkers, Ad C. van Nuenen, Neeltje A. Kootstra.

**Project administration:** Teunis B. H. Geijtenbeek.

**Resources:** Ad C. van Nuenen, Neeltje A. Kootstra, Teunis B. H. Geijtenbeek.

**Supervision:** Marta Bermejo-Jambrina, Teunis B. H. Geijtenbeek.

**Validation:** Lieve E. H. van der Donk.

**Visualization:** Lieve E. H. van der Donk, Teunis B. H. Geijtenbeek.

**Writing – original draft:** Lieve E. H. van der Donk.

**Writing – review & editing:** Lieve E. H. van der Donk, Marta Bermejo-Jambrina, Neeltje A. Kootstra, Teunis B. H. Geijtenbeek.

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
