## [Decision Letter · Decision Letter 0]

7 Aug 2023

Dear Prof. Geijtenbeek,

Thank you very much for submitting your manuscript "SARS-CoV-2 suppresses TLR4-induced immunity by dendritic cells via C-type lectin receptor DC-SIGN" for consideration at PLOS Pathogens. As with all papers reviewed by the journal, your manuscript was reviewed by members of the editorial board and by several independent reviewers. In light of the reviews (below this email), we would like to invite the resubmission of a significantly-revised version that takes into account the reviewers' comments.

We cannot make any decision about publication until we have seen the revised manuscript and your response to the reviewers' comments. Your revised manuscript is also likely to be sent to reviewers for further evaluation.

Sincerely,

Jacob S. Yount

Academic Editor

PLOS Pathogens

Alexander Gorbalenya

Section Editor

PLOS Pathogens

Kasturi Haldar

Editor-in-Chief

PLOS Pathogens

orcid.org/0000-0001-5065-158X

Michael Malim

Editor-in-Chief

PLOS Pathogens

orcid.org/0000-0002-7699-2064

Reviewer's Responses to Questions

**Part I - Summary**

Reviewer #1: In this study, van de Donk and colleagues obtained several lines of evidence that strongly suggest that SARS-CoV-2 suppresses TLR4, the major bacterial TLR on human DCs and that SARS-CoV-2 interacts with DC-SIGN leading to Raf-1-mediated suppression of TLR4 signaling crucial in the defense against bacterial infections.

The paper is well written, data well oresented, and most importantly interesting.

Reviewer #2: In this manuscript, van der Donk et al. investigate dendritic cell (DC) function after SARS-CoV-2 exposure, following up on their earlier study (van der Donk et al. (2022) Eur. J. Immunol.) showing that SARS-CoV-2 does not activate extracellular TLRs or induce DC maturation. Here, they first show that dendritic cell exposure to SARS-CoV-2 spike protein dampens responsiveness to lipopolysaccharide (LPS), a TLR4 agonist, but not the response to TLR2/6 or TLR3 agonists. After confirming previous literature that the SARS-CoV-2 spike protein binds to the C-type lectin receptor DC-SIGN, they show that blocking DC-SIGN on dendritic cells appears to restore the response to LPS in DCs pre-exposed to SARS-CoV-2 spike protein or authentic virus. Finally, Raf-1 inhibition similarly restored some of the responsiveness to LPS in DCs pre-exposed to SARS-CoV-2, supporting earlier findings where activation of DC-SIGN by bacterial or viral ligands was shown to modulate TLR4 signaling via Raf-1 (e.g. Gringuis et al. (2009) Nat. Immunol.).

Overall, the manuscript presents interesting data and addresses an important topic (bacterial co-infection in the context of COVID-19). The study is carefully executed and the data are interpreted appropriately. However, it is a fairly minor conceptual advance compared to previous work. In my opinion, further mechanistic studies (see comment 1 below) would make the manuscript more suitable for publication in PLoS Pathogens.

**Part II – Major Issues: Key Experiments Required for Acceptance**

Reviewer #1: (No Response)

Reviewer #2: 1. In the previous study by this group, it was shown that stimulation of DC-SIGN with HIV-1 enhanced LPS-induced cytokine expression (e.g. Gringuis et al. (2009) Nat. Immunol.), whereas here, SARS-CoV-2 dampens the LPS-induced cytokine response. What underlies this difference?

2. In Figure 3, the authors evaluate IL-8 expression in 293T/TLR4 cells, while IFN-beta, IL-6, IP-10, IL-10, etc. (and not IL-8) are evaluated in other figures. It would be good to also confirm in Figure 3 that the levels of one of the cytokines (e.g., IP-10, IL-6, IL-10) that exhibit decreased responsiveness to LPS after SARS-CoV-2 exposure in DCs is not decreased in the 293T/TLR4 cell model, to strengthen the conclusion that indeed the effect seen in DCs is not due to a direct interaction between SARS-CoV-2 and TLR4.

3. In Figure 4A, would recommend including an isotype control antibody.

**Part III – Minor Issues: Editorial and Data Presentation Modifications**

Reviewer #1: (No Response)

Reviewer #2: 1. Related to Figure 6, it is appreciated that the statistical analysis is clear in the figures as well as often noted in the text. The donor variance is commented upon. Even so, it is important to note in lines 136-138 that although cytokines levels appear restored with anti-DC-SIGN antibody treatment, this is only significant for IP10 and IL-6. The same goes for lines 140-143 regarding the effect of Raf-1 inhibition.

PLOS authors have the option to publish the peer review history of their article (what does this mean?). If published, this will include your full peer review and any attached files.

Reviewer #1: No

Reviewer #2: No
---

## [Editor Report · Decision Letter 1]

2 Oct 2023

Dear Prof. Geijtenbeek,

We are pleased to inform you that your manuscript 'SARS-CoV-2 suppresses TLR4-induced immunity by dendritic cells via C-type lectin receptor DC-SIGN' has been provisionally accepted for publication in PLOS Pathogens.

Best regards,

Jacob S. Yount

Academic Editor

PLOS Pathogens

Alexander Gorbalenya

Section Editor

PLOS Pathogens

Kasturi Haldar

Editor-in-Chief

PLOS Pathogens

orcid.org/0000-0001-5065-158X

Michael Malim

Editor-in-Chief

PLOS Pathogens

orcid.org/0000-0002-7699-2064
---

## [Editor Report · Acceptance letter]

12 Oct 2023

Dear Prof. Geijtenbeek,

We are delighted to inform you that your manuscript, "SARS-CoV-2 suppresses TLR4-induced immunity by dendritic cells via C-type lectin receptor DC-SIGN," has been formally accepted for publication in PLOS Pathogens.

Best regards,

Kasturi Haldar

Editor-in-Chief

PLOS Pathogens

orcid.org/0000-0001-5065-158X

Michael Malim

Editor-in-Chief

PLOS Pathogens

orcid.org/0000-0002-7699-2064